# Semi-supervised Image-to-Image translation for robust image registration

**Henrik Skibbe**[1]                   HENRIK.SKIBBE@RIKEN.JP

**Akiya Watakabe**[2]                  AKIYA.WATAKABE@RIKEN.JP

**Muhammad Febrian Rachmadi**[1]          FEBRIAN.RACHMADI@RIKEN.JP

**Carlos Enrique Gutierrez**[3]             CARLOS.GUTIERREZ@OIST.JP

**Ken Nakae**[4]                 NAKAE-K@SYS.I.KYOTO-U.AC.JP

**Hiromichi Tsukada**[3,5]             TSUKADA@ISC.CHUBU.AC.JP

**Tetsuo Yamamori**[2]              TETSUO.YAMAMORI@RIKEN.JP

[1] *Brain Image Analysis Unit, RIKEN Center for Brain Science, Wako, Japan.*

[2] *Molecular Analysis of Higher Brain Function, RIKEN Center for Brain Science, Wako, Japan.*

[3] *Okinawa Institute of Science and Technology Graduate University, Okinawa, Japan.*

[4] *Department of Systems Science, Kyoto University, Kyoto, Japan*

[5] *Center of Mathematics for Artificial Intelligence and Data Science, Chubu University, Japan*

**Editors:** Under Review for MIDL 2021

## Abstract

The Japan Brain/MINDS Project aims at studying the neural networks controlling higher brain functions in the marmoset. As part of it, we develop an image processing pipeline for marmoset brain imaging data, where various microscopy images of different modalities need to be co-registered. In initial experiments, multi-modal image registration frequently failed due to an erroneous initialization. Our data set includes images of Nissl stained brain sections, backlit images as well as images of neural tracer injections using two-photon microscopy. More than 10000 high-resolution 2D images required co-registration, a large amount that demands a reliable automation process. We implemented a semi-supervised image-to-image translation which allowed a robust image alignment initialization. With such an initial alignment, all images can be successfully registered using a state-of-the-art multi-modal image registration algorithm.

**Keywords:** Semi-supervised image-to-image translation, GAN, Image registration

## 1. Introduction

Mapping the brain structure is one important step towards understanding higher-order brain function. The Japan Brain/MINDS Project aims at studying higher brain functions in the marmoset. In this context, serial 2-photon (2p) tomography has been used to image neural connections of entire marmoset brains. For about 10% of the same brain sections, backlit and Nissl images were acquired using a second microscope. An important goal of the project is to integrate all data into a common brain image space, a critical prerequisite for group studies. For that purpose, we have developed an automated pipeline for the integration of Nissl/backlit images into the 2p reference image space. A key part of it is image registration. Figure 1 (a) illustrates our baseline image registration implementation based on ANTS (Avants et al., 2011): By registering the 2p-image (ii) to the population average template (i), the backlit image (iii) to the 2p-image, and the Nissl (iv) to the backlit image, we can

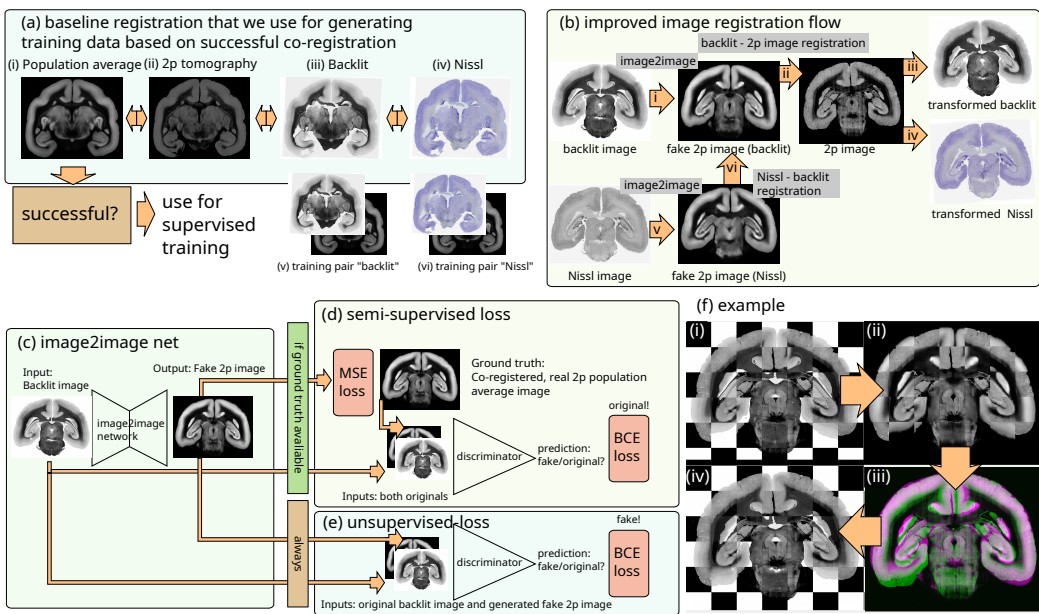

Figure 1: (a,b): Baseline and improved image registration pipeline. (c-e):Training incorporates a semi-supervised loss for paired data, and a unsupervised loss for data without ground truth. (f):The right image shows a registered image.

map - by concatenation - between all four modalities. The challenging large amount of data demands reliable automation. The image registration worked fine for most images, but frequently failed for a small percentage of them. That small percentage was causing hundreds of registrations to fail, too many to be identified and corrected manually. It turned out that a bad initial alignment for scale and orientation made the difference; a problem that we could attribute to the different imaging modalities. We tried to adjust contrast/color automatically using heuristics, but could not find a solution that "just worked". Hence we boosted the image registration process with a semi-supervised domain adaptation to alter the appearance of the Nissl and backlit images in such a way so that they appeared similar to 2p images. After this adaptation, our pipeline was able to successfully process all images of our 52 individuals in a fully automated manner. The revised registration pipeline is illustrated in Figure 1 (b). In contrast to the baseline, we first adapted the modalities of backlit (i) and Nissl (v) before registration. It is worth noting that we only used the adapted images for initial affine registration. After that, we used the original, unaltered images to fine-tune the registration with a deformable registration based on mutual information as a metric.

## 2. Proposed Approach

We improved the reliability of the image registration by using a semi-supervised image2image (i2i) translation; similar to the GAN network in (Isola et al., 2017). However, unlike other

image generation methods that aim at generating a variety of new outputs, or try to preserve individual image characteristics, we aimed at ironing out modality-specific features that can disturb image registration, like individual tracer signals. Therefore we took the population average 2p image as target domain rather than individual 2p images. For the i2i translation, we used a U-Net (Ronneberger et al., 2015) without skip connections; see Figure 1 (c). For the training, we used the imaging data from 25 brains. Images from 10 brains that could be correctly aligned with the baseline registration process were used to generate image pairs for training the i2i translation in a supervised manner; see Figure 1 (a). All remaining images were used to support training in an unsupervised manner. In GAN, the discriminator differentiates between generated images and "real" images. For samples where a ground truth image pair was not available, the discriminator loss was computed ( Figure 1 (e)). For paired images, however, the mean-squared error between the generated image and the real 2p image was added (Figure 1 (d)). An observed that we obtained better results when feeding both the input image and generated/real image into the discriminator rather than just the generated/real image. It is worth noting that image registration performance is not addressed in this paper. When initialization of the registration method is successful, there is no difference between the baseline version and the image2image boosted version. For completeness, Figure 1 (f) shows an example before alignment (i) in a checkerboard view (backlit and 2p image), (ii) + (iii) after image translation, and (iv) after alignment.

## 3. Results and Conclusion

"Conventional" image registration methods are still dominating many biomedical applications, where the outcome can have a significant impact on conclusions and decision making. A hybrid method, as proposed here, can significantly improve automation of image registration pipelines. Future steps include the extension of our results to 3D.

## Acknowledgments

This research was supported by the program for Brain Mapping by Integrated Neurotechnologies for Disease Studies (Brain/MINDS) from the Japan Agency for Medical Research and Development AMED (JP21dm0207001).

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
