# OpenReview forum: "Semi-supervised Image-to-Image translation for robust image registration"
_MIDL.io/2021/Conference/Short — MIDL 2021 Poster_

### Official Review · Reviewer_F7SA · 2021-04-30

**Confidence:** 3
**Final Rating:** 3

**Summary:**

This paper presents a pipeline for affine prealignment for 2p-nissl and 2p-backlit registration. In the current version, the pipeline is restricted to 2d slices. The nissl and backlit image is first transformed into a synthetic 2p image using a GAN and then registered to the 2p image. Using the presented pipeline, the prealignment is successful for all individuals.

**Strengths:**

- This paper tackles the important problem of erroneous prealignment (affine alignment) for multi-modal image registration
- The authors following an interesting idea that it might be more advantageous to aim for a synthetic image without modality-specific features which might disturb the image registration. Therefore, they use the population average 2p image as target domain rather than the individual 2p image.
- With the proposed image registration pipeline the prealignment is successful for all 52 individuals.
- Code for the training procedure is online (however, not the trained models)
- Data will be made publicly available in 2021/ 2022


**Weaknesses:**

- The paper is not well structured:
  - The introduction gives not only the background of the research but also the results and explains parts of the method. This might be useful for long papers but not for a short one. Especially, because in the results section no results are presented – you find the results in the end of the introduction (“After this adaption, our pipeline was able to successfully process all images of our 52 individuals in a fully automated manner”
  - How do the dataset and the training split look like? There are >10 000 2D images (abstract), 52 individuals (introduction), 25 brains in the training dataset with 10 brains with good prealignment (method)
  - The method section doesn’t explain the improved registration flow well – what do i, ii, iii, iv etc mean? Is it the order of the steps? If so, how can the transformed Nissl (iv) be generated before the generation of the fake 2p image (Nissl) (v) and the Nissl – backlit registration (vi) ?
- The presentation of the results is weak. As mentioned above, they should be presented in the result section. Furthermore, one sentence that it worked for all cases feels unsatisfactory. Moreover, it is not clear in how many cases the baseline failed.
- Missing references, e.g. Wolterink, J. M., Dinkla, A. M., Savenije, M. H., Seevinck, P. R., van den Berg, C. A., & Išgum, I. (2017, September). Deep MR to CT synthesis using unpaired data. In International workshop on simulation and synthesis in medical imaging (pp. 14-23). Springer, Cham


**Deanonymize Review:**

yes

**Detailed Comments:**

- Figure 1: f) Which right image? There are four images shown in f) and two of them are on the right.

 The following questions came up to my mind while reading this paper. They don’t have to be addressed in the current short paper, however, might be interesting to answer in an extended version of this work:
- Why do you use a “registration chain” and not registering all images to one modality (e.g. 2p)
- Have you already evaluated the impact of using the population average 2p image rather than the individual 2p image? If so, how much do you gain with it?
- How many of the failure cases of the baseline are caused due to 2d alignment? Do the prealignment and the following deformable registration also fail using 3d images?
- How are the results if the generation of synthetic images is performed in 2d but the registration is performed in 3d?
- How are the performance differences of the GAN depending on the used data - training it only unsupervised, mixed-supervised or fully semi-supervised? Would it be worth it to align the 15 image pairs manually to have a ground-truth for all image pairs?

Grammar and Spelling:
- (c-e): Training/ (f): The right (missing space)
- Therefore, we took the population average 2p image as target domain rather than […]


**Justification Of The Rating:**

Even though the presentation of the method and the results are not finalized yet, I like the presented idea and therefore, I think it is worth it to be presented at MIDL 2021. I hope that until MIDL a few more experiments might be performed and can be discussed on the poster.

**Paper Type:**

validation/application paper

**Special Issue:**

no

---

### Official Review · Reviewer_iJHr · 2021-05-01

**Confidence:** 4
**Final Rating:** 3

**Summary:**

The paper proposes a semi-supervised image-to-image translation pipeline for the robust multi-modality image alignment initialization. The method translates the appearance of Nissl and backlit images to that of 2-photon (2p) images for initial affine registration. After the initialization, they use original and real images for fine-tuned registration. Experiments show that the initialization improves the success rate of the following tasks. The hybrid method can improve efficiency and reduce the unreliability of generating methods.


**Strengths:**

The proposed method has the potential to be easily applied to other multi-modality registration tasks with few adaptations. And the semi-supervised setting fits most of the existing data sets with only partial ground truth. The image processing is more stable as the image style translation is only used for initialization and will not affect the subsequent tasks.


**Weaknesses:**

The application of image translation on the multi-modality image registration is not novel [1, 2], and many previous works deal with deformable registration, while in this paper, it’s only for affine registration.

Secondly, the experimental results are not quantitative, we can find terms like “small percentage” which is not rigorous enough.

[1] Mahapatra, Dwarikanath, et al. "Deformable medical image registration using generative adversarial networks." 2018 IEEE 15th International Symposium on Biomedical Imaging (ISBI 2018). IEEE, 2018.

[2] Arar, Moab, et al. "Unsupervised multi-modal image registration via geometry preserving image-to-image translation." Proceedings of the IEEE/CVF conference on computer vision and pattern recognition. 2020.


**Deanonymize Review:**

no

**Detailed Comments:**

1. The figures are not clear enough, the position and meaning of the arrow shape are messy.

2. The illustration of Figure 1 (f) at the end of Section 2 seems wrong, the subfigure (iii) should be (i) + (ii) if I understand correctly.


**Justification Of The Rating:**

Although the proposed method itself is not entirely novel, the application of image translation for initial medical image registration is inspiring and has the potential to improve the performance of other existing cross-modality registration methods. And the hybrid scheme involving both generative models and conventional algorithms can be widely adopted in scenarios that require high reliability as well as high accuracy.

**Paper Type:**

both

**Special Issue:**

no

---

### Meta-Review · Area_Chair_78zM · 2021-05-07

**Recommendation:** Accept (Poster)
**Confidence:** 5

**Metareview:**

Both reviewers agree that the paper has enough merits to be published at MIDL, despite a few shortcomings in presentation and limited novelty.

---

### Decision · Program_Chairs · 2021-05-11

Accept (Poster)